# Interactions in psychosocial interventions in dementia care: A systematic review protocol

**Ivy Meihua Su, Keyu Li, Winsy Wing Sze Wong** [ORCID] *

Department of Language Science and Technology, Faculty of Humanities, The Hong Kong Polytechnic University, Hong Kong SAR, China

* winsyws.wong@polyu.edu.hk

## Abstract

### Background

Psychosocial interventions are regarded as preferable in dementia care, as they have been tested as effective in preserving people living with dementia's function, reducing stress, and enhancing well-being. Typically delivered in structural activities, these interventions feature psychosocial, environmental, and behavioral interactions with people living with dementia. The degree and patterns of people living with dementia's interactional behaviors, therefore, indicate their engagement and participation in the intervention and may have an impact on the effect of psychosocial therapies.

### Objective

The systematic review aims to 1) synthesize the empirical evidence of measurement tools used to measure interactions in psychosocial interventions across various therapeutic settings and 2) investigate whether existing studies have investigated the relationship between measures of interaction and treatment outcomes, if yes, further investigation will be carried out to describe the relationship between measures and outcomes. Empirical studies and psychometric studies of scales are eligible for inclusion.

### Methods

The study protocol has been registered at the Campbell Systematic Reviews (cl2.20250120). Search will be conducted on PubMed, PsycINFO (ProQuest), Med-Line (EbscoHost), CINAHL (EbscoHost), and Cochrane Library from inception until December 31, 2025, for studies measuring people living with dementia's interactions in psychosocial interventions. The inclusion of studies will involve two independent reviewers through a two-phase procedure. During the first phase, reviewers will assess titles and abstracts, which will be followed by reading the full texts employing the predefined eligibility criteria. Data extracted will include study nature/

**Data availability statement:** No datasets were generated or analysed during the current study. All relevant data from this study will be made available upon study completion.

**Funding:** This protocol is supported by the Health and Research Medical Fund of the Health Bureau of the Government of the Hong Kong Special Administrative Region of the

People's Republic of China (reference number: 22231471). The funders had no role in study design, data collection and analysis, decision to publish, or preparation of the manuscript.

**Competing interests:** The authors have declared that no competing interests exist.

characteristics, aspects of interactions measured, measurement methods, and the relation to outcomes. The risk of bias will be assessed using the Quality Assessment Tool for Quantitative Studies for quantitative studies and the Critical Appraisal Skills Programme (CASP) checklist for qualitative studies. Results will be synthesized into a descriptive analysis, given the results of the included literature.

## Importance

This systematic review will enable a comprehensive analysis of existing frameworks used to measure interactions in psychosocial interventions across diverse settings. It may contribute to the effective measurement of interactions and provide insights into the quantitative analysis of the relation between interaction and outcome measures.

## Introduction

Dementia is an umbrella term for progressive neurocognitive disorders that impair memory, thinking, language, reasoning, and other cognitive abilities. The progressiveness of cognition decline will further lead to impaired daily functioning and independence and decreased quality of life (QoL). Apart from the negative impact on people living with dementia's individuality and life, dementia takes a toll on caregivers by placing care burdens on them. The World Health Organization (WHO) classifies it as a public health priority, considering its global prevalence and profound impact on people living with dementia and caregivers [1]. Globally, it affects over 55 million people and witnesses 10 million new cases each year, with the figure estimated to hit 139 million by 2050 as populations age [2]. Notably, while age-specific dementia incidence has declined in some Western countries due to improvements in healthcare and lifestyle, the absolute number of people living with dementia continues to rise [3]. Among the modifiable risk factors identified by the Lancet Commission, social isolation in later life is a significant contributor to dementia risk [3]. This highlights the critical importance of fostering social connectedness as both a preventive and therapeutic strategy.

Existing interventions for people living with dementia can be divided into pharmacological and nonpharmacological ones. Pharmacological treatment arguably demonstrates marginal efficacy (e.g., 1.5-ADAS-Cog-point improvement in cognitive function with donepezil) [4], significant adverse effects such as somnolence, strokes, falls, and pneumonia [5,6], which are strong predictors of poor QoL [7,8]. The minimal benefits and the variety of potential adverse effects highlight the urgent need for nonpharmacological interventions in dementia treatment [9].

Within the broad spectrum of nonpharmacological approaches, psychosocial interventions play a pivotal role. These interventions shift the focus of treatment from neurochemical pathways to psychosocial, social, and behavioral strategies to enhance well-being, preserve function, and reduce distress. Such interventions are characterized by structured routines (individualized or group-based), where emotional regulation, person-centered care, and people living with dementia's engagement are

prioritized [10]. The effects of psychosocial interventions have been proven with evidence. Specifically, sustained effects have been found in cognition [3,11,12], emotion and psychological well-being [13,14], behavior symptom management [10,15], social engagement and quality of life [11,12,16].

According to Cohen-Mansfield et al., people living with dementia's engagement in constructive and meaningful activities underpins the possibility of these interventions taking effect [17]. They investigated the mechanism of engagement and proposed that engagement is affected by environmental attributes, person attributes, and stimulus attributes, as well as the interactions between these factors [17]. In this sense, interaction facilitates engagement, emphasizing the dynamics between these factors. Similarly, Blumer discussed the notion of interaction and suggested that it is a dynamic process where individuals interpret and assign meaning to each other's actions through symbols (e.g., language, gestures) [18]. The reciprocal process of interaction is measurable by capturing eye movements, facial expressions, verbal feedback, and physical movements [17]. In dementia care, person-centered interaction is prioritized, which upholds the personhood of people living with dementia and promotes empathy, respect, and emotional connection [19]. Integrating the above interpretations of interaction, interventions can be categorized by person-environment interaction, person-caregiver interaction, person-stimulus interaction, and group-based interaction.

Person-environment interactions, including sensory stimulations, modify the physical or sensory environment to reduce agitation. Whear et al. [20] reviewed that spending time in the garden reduces levels of agitation among care home residents, and Van Maanen et al. [21] found that bright light therapy is effective for sleeping problems. Similarly, the more dynamic type—the interaction between people living with dementia and caregivers elicits holistic benefits. For example, individual reminiscence therapy, which fosters trust and emotional connection between people living with dementia and the caregiver, reported benefits in mood, well-being, and cognitive functions [22]. Stimulus-driven interactions feature a variety of triggers, including music, art, social robots, and animals, during which people living with dementia are expected to make meaningful responses to triggers. Such interactions can be sensory (e.g., animals and social robots) and cognitive (e.g., music), contributing to improved behavioral and psychologic symptoms [23], decreased stress and anxiety [24,25], and improved quality of life [25]. The last type of intervention, group-based interaction (e.g., cognitive stimulation therapy), provides a platform where people living with dementia can communicate with fellow participants or caregivers. Group discussions and synchronized movements in these interventions promote social cohesion and shared experiences in structured settings (Spector et al., 2003), improve social skills, reduce loneliness [12], and stimulate collaborative cognition [10].

The communicative feature of these interactions is beneficial to dementia treatment. Communication problems facing people living with dementia manifested as expressive language deficits (e.g., anomia) [26], receptive language challenges (e.g., impaired comprehension) [27], impaired pragmatic and social communication ability [28], repetitive speech [29], and nonverbal communication declines [30]. The variety of psychosocial therapies provides multimodal interactions for people living with dementia to maintain or improve their communication ability. For example, person-environment and stimulus-driven interactions elicit nonverbal communication by changing physical environments (e.g., light) or providing tangible objects. Moreover, group-based interactions can inspire social communication and language comprehension. Involving in interactions with the environment, stimulus, peers, and caregivers, people living with dementia are encouraged to express themselves verbally and nonverbally, as well as understand peers and caregivers.

Apart from the benefits in communication problems, interactions bear the potential to improve dementia care outcomes. Meaningful interactions can promote people living with dementia's psychological well-being [31,32] as well as physical well-being [33,34]. Specifically, to evaluate the effects of social interaction in a cognitive training program, they were randomly assigned to a high social interaction (HSI) group and a low social interaction (LSI) group while receiving cognitive training [34]. Their study suggests that social interaction (i.e., open-floor communication with experimenters) is positively related to people living with dementia's cognitive outcomes and quality of life (QoL). However, the settings of interactions investigated were in general dementia care, focusing on the attitudes and communications of nursing staff. Moreover,

existing research on the efficacy of interactions was primarily based on qualitative analysis (i.e., interviews of nursing staff), lacking statistical reasoning of the relation between interactions and outcomes. Therefore, there is a dearth of studies on interactions taking into account the mutual performances of caregivers and care recipients (i.e., people living with dementia) in therapeutic settings and conducting quantitative analysis on the relation between interactions and treatment outcomes.

Previous studies measuring interactions in dementia care focus on communications between people living with dementia, caregivers, and healthcare professionals [35–37]. Specifically, Williams et al. [37] developed an observation scale tackling the communication problems between people living with dementia and their caregivers. According to their work, the Verbal and Nonverbal Interaction Scale-CR (VNVIS-CR) takes into account caregivers' verbal and nonverbal communicative behaviors and categorizing these behaviors into sociable and unsociable types. It provides a holistic guide to measuring caregivers' behaviors but oversimplifies people living with dementia's communicative performance, as verbal outputs are measured in terms of frequencies without further classification. Comparably, Mabire et al. [36] paid more attention to people living with dementia's interaction behaviors and developed a multimodal scale to observe how they communicate with residents as well as care staff from visual, verbal, and facial perspectives. Meanwhile, interaction measures in structured psychosocial activities vary across different settings. Therapies including robots and animals as stimuli have spotlighted both people living with dementia's verbal and nonverbal performance [38–40] while interventions concerning music [41], communication [42], and group-based activities [43] paid more attention to their verbal outputs. The majority of interaction measures have been examined quantitatively [38–42], where frequencies and durations of their performance were recorded. Apart from quantitative studies, Orfanos et al. [44] probed into interactions in group-based cognitive stimulation therapy qualitatively, which revealed participants' perception of interaction.

Given the diversity of psychosocial interventions, interaction measures have taken different focuses to align with different treatment goals. A systematic synthesis of existing interaction measures can facilitate a comprehensive understanding of interactions in dementia care and the efficacy of various approaches. To have a better understanding of interactions in a wide range of therapies, this study aims to give a systematic review of the existing interaction measures applied in empirical studies and tackle the following questions:

1. What are the nature and characteristics of the studies in which interaction with people living with dementia was measured (e.g., type of intervention, characteristics of participants)?

2. Which aspects of interactions are measured (e.g., verbal or nonverbal)?

3. How were interactions measured in different therapeutic settings (e.g., parameters; qualitative or quantitative in nature; linguistics or pragmatic; coding systems; framework/analytic methods used)?

4. Did these studies try to investigate the relation between measures and outcomes? If yes, further investigation will be carried out to describe the relationship between measures and outcomes.

## Methods

This paper will examine the existing research on psychosocial interventions for people living with dementia where participants' interactions in corresponding therapeutic settings were measured. A comprehensive review will be conducted to include quantitative studies and qualitative studies with empirical evidence. The review follows the Preferred Reporting Items for Systematic Reviews and Meta-analyses (PRISMA) guidelines (S1 File) [45]. The study protocol has been registered at the Campbell Systematic Reviews (cl2.20250120).

### Eligibility criteria

**Type of participants.** People living with dementia of any type, including Alzheimer's Disease, dementia with Lewy bodies, vascular dementia, alcohol-related dementia, Parkinson's disease dementia, and primary progressive aphasia, are

of interest. The dementia can be mild, moderate, or severe. Studies involving caregivers and facilitators in relation to the targeted population are eligible for inclusion. Instead, studies involving a mixed population will be considered only if 80% of the population consists of people living with dementia.

**Types of intervention.** Studies will be included if they describe methods to measure interactions concerning people living with dementia in psychosocial interventions. Examples of these interventions include music therapy, robot therapy, animal therapy, cognitive stimulation therapy, and reminiscence therapy, to name a few. The interactions can be Person-Environment, Person-Caregiver, Person-Stimulus, or Group-Based ones. Studies will be excluded if pharmacological approaches or purely physiological interventions (e.g., electrical stimulation, dietary changes) are employed or if the interactional process is not measured.

**Types of study.** Empirical studies (RCTs, non-RCTs, observational studies, pre- and post- studies, and qualitative studies) and psychometric studies of scales are eligible for inclusion. Exclusion will be applied to review articles and scoping/systematic reviews, theoretical papers, editorials, and opinion pieces to ensure the review focuses on applied measurement tools. No restrictions on publication language will be imposed. Studies in languages not fluent among the author team will undergo initial machine translation (i.e., Google Translate) for screening. For all studies included in the final review, key data (e.g., outcomes, methodology, results) will be translated by a fluent academic colleague or a professional translation service to guarantee accuracy. We will contact the original authors, where feasible, to verify critical data interpretations.

**Types of outcome measures.** Measurement of interaction processes (e.g., observation scale, coding system, quantitative analysis of interaction) among people living with dementia and the environment/caregivers/peers/stimuli/facilitators during the intervention will be the primary outcome. The outcomes of this systematic review will include a summary of indicators used to capture their interactional behaviors in psychosocial settings.

**Other exclusion criteria.** Studies that are not published after December 31, 2025, will not be included.

### Search strategy

**Electronic searches.** An initial search will be conducted on databases: PubMed, PsycINFO (ProQuest), MedLine (EbscoHost), CINAHL (EbscoHost), Cochrane Library. The development of the search terms focus on three key concepts: the population (people with dementia or cognitive impairment), the intervention type (nonpharmacological interventions), and the outcomes of interest (interaction). The full list of search terms is detailed in Appendix 1 (S2 File). Boolean operators (AND) will be applied.

**Searching other sources.** Supplementary Search will be conducted by identifying grey literature. We will search ProQuest Dissertations and Theses Global and Google Scholar (screening the first 200 results) using a simplified combination of the key search terms (e.g., "dementia" AND "interaction" AND "nonpharmacological"). We will also screen the proceedings of major conferences (e.g., Alzheimer's Association International Conference) from the last three years.

Citation tracking will be performed by manually screening the reference lists of all studies ultimately included in the review (backward citation tracking) to identify further relevant research.

Finally, systematic reviews identified during the database search will be flagged. We will screen the reference lists of these relevant systematic reviews to identify potentially eligible primary studies. Any primary studies identified through citation tracking or systematic reviews will be subject to the same two-phase screening process (title/abstract and full text) as the database search results.

**Screening and inclusion.** Searched records from all sources will be combined and uploaded to Covidence, where duplicates will be removed. An inter-rater reliability of 90% will be assured before conducting a two-stage screening. In the first stage, two independent reviewers (Su and Li) will screen the titles and abstracts of all identified records against the eligibility criteria. During this process, blind voting will be used to minimise bias. In the second phase, two researchers will then continue to read the full text of the preliminary-selected records. Disagreements will be resolved through discussion or by a third reviewer (author Wong).

## Data extraction

A pilot data extraction will be conducted to develop a standardized table using an Excel spreadsheet. Two reviewers will independently extract data based on the standardized sheet.

In response to the first research question (the nature and characteristics of studies), the following data will be extracted:

• Study characteristics: author, year, country, design, setting, funding

• Participant characteristics: number, dementia type/severity, demographics

• Intervention details: type, duration, frequency, facilitator, specific activities

To answer the second question (which aspect of interactions were measured), verbal or nonverbal perspective will be coded.

The third question (how interactions were measured) will be resolved by extracting information on interaction measurement: specific tool/method used, parameters (e.g., frequency, duration, content, quality), nature (qualitative/quantitative), coding system/framework.

For the last question, which explores whether the relation between interaction measures and outcomes was investigated, any reported outcomes (cognitive, behavioral, QoL, mood, engagement, etc.) and tools used will be extracted.

## Critical appraisal

The quality of studies will be assessed using relevant checklists. The quality of quantitative studies will be assessed using Joanna Briggs Institute (JBI) Critical Appraisal Tools [46]. Specifically, we will use the JBI Checklist for Randomized Controlled Trials for RCTs, the JBI Checklist for Quasi-Experimental Studies for non-randomized experimental studies, and the JBI Checklist for Analytical Cross-Sectional Studies for observational designs [46]. Key methodological domains, including randomization, blinding, and statistical analysis, will be assessed with these tools. For studies focusing specifically on the psychometric properties of interaction scales, we will utilize the COSMIN Risk of Bias checklist to evaluate validity and reliability [47]. For qualitative studies, the Critical Appraisal Skills Programme (CASP) checklist [48], where 10 questions were designed to assist reviewers in assessing the reliability, objectivity, and rigor of qualitative research will be employed.

## Data synthesis and analysis

Given anticipated heterogeneity (in populations, interventions, measures), a narrative synthesis approach will be employed following the Synthesis Without Meta-analysis (SWiM) reporting guidelines [49]:

**Grouping and tabulation.** Studies will be grouped by the type of psychosocial intervention (e.g., robot therapy vs. music therapy) and the aspect of interaction measured (verbal vs. nonverbal). Findings will be presented in standardized summary of findings tables.

**Narrative synthesis.** The synthesis will be structured around the four review questions:

Q1. Study nature/characteristics: descriptive statistics (number of studies, percentages) and narrative summary will be presented in tables and text (intervention types, participant demographics/dementia severity, settings, geographical distribution).

Q2. Aspects of interaction measured: studies will be categorized by the primary aspect of interaction measured (e.g., verbal, nonverbal).

Q3. Measurement methods: studies will be analyzed and compared based on: parameters (e.g., frequency, duration, content, sequence), nature (quantitative, qualitative, mixed methods), focus (linguistic, pragmatic, behavioral), and specific coding systems/analytic frameworks used.

Q4. Relation to outcomes: evidence regarding associations between interaction measures and study outcomes (cognitive, behavioral, QoL, mood, engagement) will be synthesized narratively. Reported correlations, regression results, or qualitative themes linking interaction to outcomes will be tabulated and described.

**Assessment of heterogeneity and bias.** Heterogeneity will be explored narratively by comparing findings across different study designs (e.g., RCTs vs. observational) and settings. The potential of interaction measures to explain outcomes will be critically discussed in light of the quality assessment (JBI/COSMIN), giving greater weight to findings from studies with lower risk of bias.

## Significance of the current study protocol

This systematic review will facilitate the understanding of interaction in psychosocial interventions for people living with dementia. The outcome of this study will identify interactional behaviors of people living with dementia across diverse interventions and methods of interaction measures featuring people living with dementia and the environment (e.g., care-givers, peers, robot). The statistical relation between interactions and dementia care outcomes, if there is any, will also be presented. It will help to establish a comprehensive measurement system for interactions in therapeutic settings, which will contribute to uncovering the facilitative strategies in dementia care.

## Supporting information

**S1 File. PRISMA 2020 checklist.** Completed PRISMA 2020 checklist for this systematic review protocol.
(DOCX)

**S2 File. Appendix 1. Search strategy.** Full search strategy for each database (PubMed, PsycINFO, MedLine, CINAHL, Cochrane Library).
(DOCX)

## Author contributions

**Conceptualization:** Ivy Meihua Su, Winsy Wing Sze Wong.

**Data curation:** Ivy Meihua Su, Keyu Li, Winsy Wing Sze Wong.

**Funding acquisition:** Winsy Wing Sze Wong.

**Investigation:** Ivy Meihua Su, Winsy Wing Sze Wong.

**Methodology:** Ivy Meihua Su, Winsy Wing Sze Wong.

**Project administration:** Winsy Wing Sze Wong.

**Software:** Keyu Li.

**Supervision:** Winsy Wing Sze Wong.

**Validation:** Keyu Li, Winsy Wing Sze Wong.

**Writing – original draft:** Ivy Meihua Su.

**Writing – review & editing:** Ivy Meihua Su, Winsy Wing Sze Wong.

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
