## [Decision Letter · Decision Letter 0]

19 Jan 2026

PONE-D-25-60484Interactions in dementia therapies: A systematic review protocol

PLOS One

Dear Dr. Wong,

Thank you for submitting your manuscript to PLOS ONE. After careful consideration, we feel that it has merit but does not fully meet PLOS ONE’s publication criteria as it currently stands. Therefore, we invite you to submit a revised version of the manuscript that addresses the points raised during the review process.

The manuscript has been assessed by one peer reviewer and I ask the authors to consider the points raised by the peer reviewer and also take into account my additional comments below:

1) The abstract states two objectives, with the second objective aiming at the “relation between interactions and outcomes.”, which sounds like aiming at reporting the actual results of the relationships, while in the manuscript text, the fourth objective only states the description of whether this relationship is being investigated and, if so, how, without mentioning results. Please clarify.

(2) The review has been registered with the Campbell Database. Unfortunately, I am unable to access the specified registration. Please provide a link that allows to access to the registration. Furthermore, I do not understand why, as it is aimed to be a Campbell Review, the protocol should be published there but in this journal. Please explain.

(3) Only participants with dementia will be included and studies with mixed populations excluded. Is there a cutoff, for example, 80% of people with dementia for individual studies or are all papers excluded as soon as there is just one person without dementia among participants?

(4) The timeframe for the search and for the inclusion of studies should be updated. From my perspective, there is no reason, why only studies up to April 2025 should be included.

(5) The critical appraisal instrument for quantitative studies seems appropriate here. Please select an appropriate instrument for each (quantitative) study design. It is not clear from the manuscript to what extent the quality assessment will be actually connected with the study results. Please point out. It seems inappropriate to only list the study quality of included studies.

(6) As the reviewer noted, it is not entirely clear how the screening process actually works. As I understand it from the manuscript, not all full texts are reviewed by two independent reviewers, but only 10%, and if sufficient inter-rater reliability is demonstrated, this procedure is then carried forward. So in my understanding, only one reviewer reviews a given set of full texts at a time. However, this is reported differently in the PRISMA-P document, which states that the reviewers are independently checking full texts. Please clarify

(7) The PRISMA-P checklist contains methodological steps not reported in the manuscript, such as Subgroup Analyzes, Sensitivity Analyzes, and Meta-Regression or the application of “the SWiM or Popay et al. framework” or using GRADE. Please check and synchronize.

(8) Please avoid colloquial terms like "till" instead of "until". Also, the term "myriad" seems inappropriate, please use “plenty" or something similar.

We look forward to receiving your revised manuscript.

Kind regards,

Sascha Köpke

Academic Editor

PLOS One

Journal Requirements:

“This protocol is supported by the Health and Research Medical Fund of the Health Bureau of the Government of the Hong Kong Special Administrative Region of the People’s Republic of China (reference number: 22231471) and the General Research Fund of the Research Grants Council of the Hong Kong Special Administrative Region of the People’s Republic of China (reference number: 15602024).”

“This protocol is supported by the Health and Research Medical Fund of the Health Bureau of the Government of the Hong Kong Special Administrative Region of the People’s Republic of China (reference number: 22231471).”

“This protocol is supported by the Health and Research Medical Fund of the Health Bureau of the Government of the Hong Kong Special Administrative Region of the People’s Republic of China (reference number: 22231471).”

4. When completing the data availability statement of the submission form, you indicated that you will make your data available on acceptance. We strongly recommend all authors decide on a data sharing plan before acceptance, as the process can be lengthy and hold up publication timelines. Please note that, though access restrictions are acceptable now, your entire data will need to be made freely accessible if your manuscript is accepted for publication. This policy applies to all data except where public deposition would breach compliance with the protocol approved by your research ethics board. If you are unable to adhere to our open data policy, please kindly revise your statement to explain your reasoning and we will seek the editor's input on an exemption. Please be assured that, once you have provided your new statement, the assessment of your exemption will not hold up the peer review process

Reviewers' comments:

Reviewer's Responses to Questions

**Comments to the Author**

1. Does the manuscript provide a valid rationale for the proposed study, with clearly identified and justified research questions?

Reviewer #1: Yes

2. Is the protocol technically sound and planned in a manner that will lead to a meaningful outcome and allow testing the stated hypotheses?

Reviewer #1: Partly

3. Is the methodology feasible and described in sufficient detail to allow the work to be replicable?

Reviewer #1: Yes

4. Have the authors described where all data underlying the findings will be made available when the study is complete?

Reviewer #1: Yes

5. Is the manuscript presented in an intelligible fashion and written in standard English?

Reviewer #1: Yes

6. Review Comments to the Author

You may also provide optional suggestions and comments to authors that they might find helpful in planning their study.

Reviewer #1: Dear Authors,

thank you very much for the opportunity to review your review protocol with the title "Interactions in dementia therapies: A systematic review protocol". I only have a few comments and hope the authors find these helpful.

General: Please avoid the use of an abbreviate for people living with dementia. Please see this recommendation: https://www.alzheimers.org.uk/sites/default/files/2018-09/Positive%20language%20guide_0.pdf

Please avoid the word elderly for older people. For this see this recommendation: Editor’s Message: Use of the Term “Elderly” in Journal of GERIATRIC Physical Therapy

Abstracts: Reading your definition of non-pharmacological interventions it seems you mean psychosocial interventions.

Introduction: Please take a look on the LANCET paper for risk factors for dementia and prevention programs. It is perhaps important to point out that age-related dementia is declining in Western countries. However, despite the number of older people, the number of dementia patients is increasing, and prevention could be an important lever here.

Methods: Please consider whether this is truly a systematic review or whether it is more of a scoping or integrative review.

Why didn’t you consider people with FTD?

How do you search for grey literature?

How do you conduct a citation tracking?

How do you proceed if you find a study of interested identified in a review?

Studies published after April 1, 2025 will not be included – I think you have to rephrase this sentence because if you need to conduct an update for publishing your review you will also include studies from 2025.

Was a library included in the development of the search string?

Entries will be divided into two parts? I think this sentence is misleading. Every study need to be screened by two reviewer or do you really mean that every study will be screened by one reviewer?

Don’t understand why a third reviewer is reviewing 10% of all titles.

7. PLOS authors have the option to publish the peer review history of their article (what does this mean?). If published, this will include your full peer review and any attached files.

Reviewer #1: **Yes:** Mike Rommerskirch-Manietta

---

## [Author Response · Author response to Decision Letter 1]

20 Mar 2026

Response Letter

Manuscript ID: PONE-D-25-60484R1

Title: Interactions in psychosocial interventions in dementia therapies: A systematic review protocol

Dear Ms. Trisha Mae Tañedo Perez,

Thank you for your detailed feedback and guidance on our manuscript (PONE-D-25-60484R1). We have carefully addressed all the revision requests as required, and the specific completion details are outlined below:

1. Manuscript File Inventory Arrangement

We have checked the submission file inventory in the Editorial Manager system and marked the description column of the most recent manuscript file with "latest" for clear identification, ensuring the editorial team can easily locate the valid version for review.

2. Role of the Funders Statement

The role of funder has been stated in the cover letter as instructed.

3. Manuscript Title Consistency

We have thoroughly verified the manuscript title and ensured the title displayed on the online submission form is identical to that in the manuscript document, with no discrepancies existing between the two.

4. Author Addition

We have updated the submission data via the "Edit Submission" function in the system and successfully added Ivy Meihua Su to the author list, with all relevant author information completed accurately.

Thank you again for your patience and support for our work. Should you need any further information or clarification, please do not hesitate to contact us at any time.

Sincerely,

Winsy Wing Sze Wong

---

## [Editor Report · Decision Letter 1]

26 Mar 2026

PONE-D-25-60484R1Interactions in psychosocial interventions in dementia care: A systematic review protocolPLOS One

Dear Dr. Wong,

Thank you for submitting your manuscript to PLOS ONE. After careful consideration, we feel that it has merit but does not fully meet PLOS ONE’s publication criteria as it currently stands. Therefore, we invite you to submit a revised version of the manuscript that addresses the points raised during the review process.

**ACADEMIC EDITOR:**

Before, we can assess the revision, please submit a proper point-by-point response, individually addressing the reviewers comments.

We look forward to receiving your revised manuscript.

We look forward to receiving your revised manuscript.

Kind regards,

Sascha Köpke

Academic Editor

PLOS One

Journal Requirements:

2. Include with your response a letter that responds to each point raised by the academic editor and reviewer(s) in the earlier decision letter you received. You should upload this letter as a separate file labeled 'Response to Reviewers'. For your convenience, the comments from the academic editor and reviewer are copied below

Academic editor:

1) The abstract states two objectives, with the second objective aiming at the “relation between interactions and outcomes.”, which sounds like aiming at reporting the actual results of the relationships, while in the manuscript text, the fourth objective only states the description of whether this relationship is being investigated and, if so, how, without mentioning results. Please clarify.

(2) The review has been registered with the Campbell Database. Unfortunately, I am unable to access the specified registration. Please provide a link that allows to access to the registration. Furthermore, I do not understand why, as it is aimed to be a Campbell Review, the protocol should be published there but in this journal. Please explain.

(3) Only participants with dementia will be included and studies with mixed populations excluded. Is there a cutoff, for example, 80% of people with dementia for individual studies or are all papers excluded as soon as there is just one person without dementia among participants?

(4) The timeframe for the search and for the inclusion of studies should be updated. From my perspective, there is no reason, why only studies up to April 2025 should be included.

(5) The critical appraisal instrument for quantitative studies seems appropriate here. Please select an appropriate instrument for each (quantitative) study design. It is not clear from the manuscript to what extent the quality assessment will be actually connected with the study results. Please point out. It seems inappropriate to only list the study quality of included studies.

(6) As the reviewer noted, it is not entirely clear how the screening process actually works. As I understand it from the manuscript, not all full texts are reviewed by two independent reviewers, but only 10%, and if sufficient inter-rater reliability is demonstrated, this procedure is then carried forward. So in my understanding, only one reviewer reviews a given set of full texts at a time. However, this is reported differently in the PRISMA-P document, which states that the reviewers are independently checking full texts. Please clarify

(7) The PRISMA-P checklist contains methodological steps not reported in the manuscript, such as Subgroup Analyzes, Sensitivity Analyzes, and Meta-Regression or the application of “the SWiM or Popay et al. framework” or using GRADE. Please check and synchronize.

(8) Please avoid colloquial terms like "till" instead of "until". Also, the term "myriad" seems inappropriate, please use “plenty" or something similar.

Reviewer 1:

General: Please avoid the use of an abbreviate for people living with dementia. Please see this recommendation: https://www.alzheimers.org.uk/sites/default/files/2018-09/Positive%20language%20guide_0.pdf

Please avoid the word elderly for older people. For this see this recommendation: Editor’s Message: Use of the Term “Elderly” in Journal of GERIATRIC Physical Therapy

Abstracts: Reading your definition of non-pharmacological interventions it seems you mean psychosocial interventions.

Introduction: Please take a look on the LANCET paper for risk factors for dementia and prevention programs. It is perhaps important to point out that age-related dementia is declining in Western countries. However, despite the number of older people, the number of dementia patients is increasing, and prevention could be an important lever here.

Methods: Please consider whether this is truly a systematic review or whether it is more of a scoping or integrative review.

Why didn’t you consider people with FTD?

How do you search for grey literature?

How do you conduct a citation tracking?

How do you proceed if you find a study of interested identified in a review?

Studies published after April 1, 2025 will not be included – I think you have to rephrase this sentence because if you need to conduct an update for publishing your review you will also include studies from 2025.

Was a library included in the development of the search string?

Entries will be divided into two parts? I think this sentence is misleading. Every study need to be screened by two reviewer or do you really mean that every study will be screened by one reviewer?

Don’t understand why a third reviewer is reviewing 10% of all titles."

---

## [Author Response · Author response to Decision Letter 2]

31 Mar 2026

Winsy Wing Sze Wong, Ph.D.

Corresponding Author

Department of Language Science and Technology

The Hong Kong Polytechnic University

Hung Hom, Kowloon, Hong Kong

Email: winsyws.wong@polyu.edu.hk

Phone: (852) 2766 7454

March 3, 2026

Sascha Köpke

Editorial Office

PLOS ONE

Re: Response to Reviewers for Manuscript PONE-D-25-60484

Title: Interactions in psychosocial interventions in dementia care: A systematic review protocol

Dear Dr. Köpke,

Thank you for the opportunity to revise our manuscript. We appreciate the thoughtful and constructive feedback provided by you and the reviewer. We have fully and carefully addressed all revision requirements—including the initial administrative submission requests, the academic methodological comments, the reviewer’s substantive suggestions, and the latest journal submission guidelines. All modifications have been clearly highlighted in the tracked version of the manuscript, and all authors have reviewed and approved the fully revised manuscript and all accompanying submission materials.

We confirm that all prior administrative revision requirements have been fully implemented and retained in this resubmission, and we have strictly complied with all new editorial instructions for the review process. Below is a comprehensive point-by-point response to all raised comments and revision requests.

I. Initial Administrative Revision Requirements (Fully Implemented & Retained)

1. Manuscript File Inventory Arrangement

We have checked the submission file inventory in the Editorial Manager system and marked the description column of the most recent manuscript file with "latest" for clear identification, ensuring the editorial team can easily locate the valid, up-to-date version for review; no outdated or absent files are present in the inventory.

2. Role of the Funders Statement

As instructed, the funder role statement has been clearly stated in the cover letter for this resubmission, and the updated funding disclosure has been revised in line with journal requirements (see Section IV for full statement). The core declaration remains unchanged: The funders had no role in study design, data collection and analysis, decision to publish, or preparation of the manuscript.

3. Manuscript Title Consistency

We have thoroughly verified and confirmed full consistency of the manuscript title (Interactions in psychosocial interventions in dementia care: A systematic review protocol) across the online submission form and the manuscript document, with no discrepancies existing between the two.

4. Author Addition

We have updated the submission data via the "Edit Submission" function in the Editorial Manager system and successfully added Ivy Meihua Su to the author list, with all relevant and complete author information accurately completed and retained in the system.

II. Response to Academic Editor’s Comments

1. Clarification of Objectives (Relation between interactions and outcomes) Comment: The abstract states the second objective aims at the “relation between interactions and outcomes,” which sounds like reporting actual results.

Response: We agree that the original wording was ambiguous. We have clarified in the Objective section that our aim is to investigate whether and how existing studies have examined this relationship, rather than conducting a meta-analysis of the relationship ourselves at this stage. We will synthesize the reported evidence on this relationship.

Action: The objective text in the abstract has been edited ‘investigate whether existing studies have investigated the relationship between measures of interaction and treatment outcomes, if yes, further investigation will be carried out to describe the relationship between measures and outcomes’ (p.3, lines 28-33). And research objective 4 in the main text has been refined to: “further investigation will be carried out to describe the relationship between measures and outcomes” (p. 13, lines 189-191 in the manuscript).

2. Campbell Registration and Journal Choice Comment: Please provide a link to the registration. Why publish in PLOS ONE if it is a Campbell Review?

Response: We did not intend to submit the manuscript to Campbell review. We applied Campbell Review for a title registration and were accepted and listed on their website: https://onlinelibrary.wiley.com/doi/10.1002/cl2.70067

We have chosen to publish the protocol in PLOS ONE to reach a broader, interdisciplinary audience (psychology, nursing, linguistics) that extends beyond the traditional scope of Campbell, ensuring the methodology receives wide visibility.

3. Participants and Cutoff Comment: Is there a cutoff for mixed populations (e.g., 80%)?

Response: Yes, we have established a clear cutoff. Action: We have specified in the “Eligibility criteria" section: "Studies involving a mixed population will be considered only if 80% of the population consists of people living with dementia” (p. 14, lines 207-208 in the manuscript and item 8, Participants in the PRISMA list).

4. Timeframe for Search Comment: The timeframe should be updated; there is no reason to limit to April 2025.

Response: We apologize for the confusion caused by the future date in the previous draft.

Action: We have updated the search strategy to state that the search will be conducted “from inception to December 31, 2025” (p.4, line 39 in the manuscript and items 8, 9, and 10, appendix in PRISMA list). We removed the exclusion criterion regarding the future date.

5. Critical Appraisal Instruments Comment: Please select an appropriate instrument for each design and explain how quality assessment connects to results.

Response: We have specified the tools for each design: JBI Checklist for RCTs, JBI for Quasi-Experimental, JBI for Cross-Sectional, and the COSMIN Risk of Bias checklist for psychometric studies. We also clarified that quality assessment will inform the narrative synthesis (giving greater weight to studies with lower risk of bias).

Action: The "Critical Appraisal" section has been expanded with these specific details (p.18-19, lines 290-297-2 in the manuscript and item 14 in the PRISMA list).

6. Screening Process Clarification Comment: It is unclear if all full texts are reviewed by two independent reviewers. The PRISMA-P document states independent checking.

Response: This was a phrasing error in the previous draft. We confirm that all titles/abstracts and all full texts will be screened by two independent reviewers. The mention of "10%" referred to the initial pilot testing of the screening form to ensure inter-rater reliability, not the screening itself.

Action: We have rewritten the "Screening and inclusion" section to clearly state that two independent reviewers will screen all records at both stages (p.17, lines 264-269 in the manuscript).

7. PRISMA-P Synchronization Comment: The checklist contains steps not reported (e.g., SWiM, subgroup analysis).

Response: We have synchronized the text with the checklist. We are utilizing the SWiM (Synthesis Without Meta-analysis) guidelines for the narrative synthesis.

Action: We have explicitly cited the SWiM guidelines in the "Data Synthesis" section (p.19, lines 303-310 in the manuscript, and items 15c, 16, 17 in the PRISMA list).

8. Language (Colloquialisms) Comment: Avoid "till" and "myriad."

Response: We have removed these terms.

Action: "Till" has been replaced with "until" (or removed in the context of the date update), and "myriad" has been replaced with "variety" or "diverse" (p. 4, line 38; p.4, line 51 in the manuscript)

III. Response to Reviewer #1’s Comments

1. Terminology (People living with dementia) Comment: Please avoid abbreviations for people living with dementia.

Action: We have removed the abbreviation "PwD" and replaced it with "people living with dementia" throughout the manuscript (p. 3, line 21, 24; p. 4, line 39; p. 6, line 64; p.7, line 73, 85, 90; p. 8, line 101, 109; p. 9, line 111, 114, 119, 120, 125, 129; p. 10, line 103, 135, 139, 141, 144; p. 11, line 151, 154, 156, 161, 163, 164, 167; p. 12, line 169, 171, 183; p. 13, line 194; p. 14, line 211; p. 15, line 231, 234; p. 21, line 332, 333, 334 in the manuscript).

2. Terminology (Elderly) Comment: Please avoid the word "elderly."

Action: We have replaced "elderly" with "older adults" or "older people" throughout the text, consistent with the Journal of Geriatric Physical Therapy recommendations.

3. Abstract (Intervention Definition) Comment: Definition of non-pharmacological interventions seems to mean psychosocial interventions.

Action: We have updated the title and the text to specifically refer to "psychosocial interventions" to be more precise and aligned with our inclusion criteria (p. 3, lines 20-26, lines 29; p. 4, line 40, 51; p. 7, lines 80-82, line 86, 91; p. 8, line 103, 105; p. 9, line 129; p. 11, line 166; p. 12, line 168, 175, 178; p. 13, line 193; p. 14, line 211, 215-216; p. 15, line 232, 234; p. 21, line 332, line 333, line 337 in the manuscript).

4. Introduction (Lancet/Statistics) Comment: Reference the Lancet paper regarding risk factors and prevention. Point out declining age-related incidence vs. rising absolute numbers.

Action: We have incorporated the Lancet Commission (2020) reference and added a sentence in the Introduction highlighting that while age-specific incidence may be declining in some Western countries, the absolute number of cases continues to rise, emphasizing the need for intervention (p. 6, lines 60-72 in the manuscript).

5. Review Type (Systematic vs. Scoping) Comment: Consider if this is a systematic or scoping review.

Response: We would like to clarify that this is a systematic review because we aim to critically appraise the quality of the studies (using JBI/COSMIN tools) and synthesize evidence on the measurement properties and outcomes, rather than just mapping key concepts.

Action: We have maintained the systematic review designation but ensured our methodology (critical appraisal and synthesis) supports this (p. 3, line 28; p.12, line 180; p. 13, line 196; p. 14, lines 221-223; p. 19, lines 303-310 in the manuscript and item 8, study design in the PRISMA list).

6. Inclusion of FTD Comment: Why didn’t you consider people with FTD?

Response: We do include FTD.

Action: In the "Types of participants" section, we have explicitly listed "primary progressive aphasia" (a form of FTD) and "dementia of any type," ensuring FTD is covered. Also in the search terms (provided in the Appendix 1 of PRISMA-P checklist), frontotemporal dementia is added (p. 8 of PRISMA checklist).

7. Grey Literature and Citation Tracking Comment: How do you search grey literature and conduct citation tracking?

Action: We have added details to the "Searching other sources" section, specifying that we will search ProQuest Dissertations and Theses, Google Scholar, and conference proceedings. We also clarified that we will perform backward citation tracking on included studies (p. 16, lines 246-259 in the manuscript and items 9 and 10 in the PRISMA list).

8. Search Date (April 2025) Comment: Rephrase the sentence about studies published after April 2025.

Action: As noted in the response to the Editor, we have removed the future cutoff date and stated the search covers "inception to December 31, 2025" (p.4, line 38-39; p.15, line 236 in the manuscript)

9. Screening Procedure Comment: "Entries will be divided into two parts" is misleading.

Response: We agree.

Action: We deleted the confusing sentence. The text now clearly states that two independent reviewers (Su and Li) will screen the titles/abstracts of all identified records (p. 17, lines 264-269 in the manuscript).

10. Third Reviewer Comment: Don’t understand why a third reviewer is reviewing 10% of all titles.

Response: This was a remnant of a previous draft regarding reliability checks.

Action: We have clarified that the third reviewer (Author Wong) serves as an arbitrator to resolve disagreements between the two primary reviewers (p. 17, lines 268-269 in the manuscript).

11. Library Consultation Comment: Was a library included in the development of the search string?

Response: The search strategy was developed by the research team using standard MeSH terms and keywords derived from key literature in the field. We also consulted the resources, guides, and tutorials provided by the library of our institute. (https://libguides.lb.polyu.edu.hk/syst_review/find_SR)

IV. Compliance with PLOS ONE Journal Submission & Style Requirements

1. Required Submission Files

In accordance with your latest editorial instructions, we have prepared and uploaded all required supplementary files as separate documents in the Editorial Manager’s "Submissions Needing Revision" folder, labeled exactly as specified:

• Response to Reviewers: This comprehensive document with a detailed point-by-point response to all Academic Editor and Reviewer #1 comments;

• Revised Manuscript with Track Changes: A marked-up copy of the manuscript clearly highlighting all modifications made in response to the review process;

• Manuscript: A clean, unmarked version of the revised manuscript with no track changes or editorial marks.

2. PLOS ONE Style Requirements

The manuscript fully adheres to all PLOS ONE style guidelines, including correct file naming conventions, template formatting, and editorial writing standards.

3. Funding Statement and Acknowledgments

Per journal request, all funding-related text has been removed from the Acknowledgments section of the manuscript. The updated Funding Statement (to be reflected in the online submission system) is:

This protocol is supported by the Health and Research Medical Fund of the Health Bureau of the Government of the Hong Kong Special Administrative Region of the People’s Republic of China (reference number: 22231471) and the General Research Fund of the Research Grants Council of the Hong Kong Special Administrative Region of the People’s Republic of China (reference number: 15602024). The funders had no role in study design, data collection and analysis, decision to publish, or preparation of the manuscript.

4. Data Availability

We confirm that all data underlying the findings of this systematic review will be made fully available without restriction at the time of publication. As this is a systematic review protocol, no research datasets have been generated to date.

5. Funder Role & Financial Disclosure

No changes have been made to our financial disclosure, and the core funder role statement remains consistent in the cover letter and manuscript: The funders had no role in study design, data collection and analysis, decision to publish, or preparation of the manuscript.

We believe the manuscript is now significantly improved and ready for publication.

Sincerely,

Winsy Wing Sze Wong Department of Language Science and Technology The Hong Kong Polytechnic University

---

## [Decision Letter · Decision Letter 2]

21 Apr 2026

Interactions in psychosocial interventions in dementia care: A systematic review protocol

PONE-D-25-60484R2

Dear Dr. Wong,

We’re pleased to inform you that your manuscript has been judged scientifically suitable for publication and will be formally accepted for publication once it meets all outstanding technical requirements.

Kind regards,

Sascha Köpke

Academic Editor

PLOS One

Reviewers' comments:

Reviewer's Responses to Questions

**Comments to the Author**

1. Does the manuscript provide a valid rationale for the proposed study, with clearly identified and justified research questions?

Reviewer #1: Yes

2. Is the protocol technically sound and planned in a manner that will lead to a meaningful outcome and allow testing the stated hypotheses?

Reviewer #1: Yes

3. Is the methodology feasible and described in sufficient detail to allow the work to be replicable?

Reviewer #1: Yes

4. Have the authors described where all data underlying the findings will be made available when the study is complete?

Reviewer #1: Yes

5. Is the manuscript presented in an intelligible fashion and written in standard English?

Reviewer #1: Yes

6. Review Comments to the Author

You may also provide optional suggestions and comments to authors that they might find helpful in planning their study.

Reviewer #1: Thank you very much for your detailed response. I wish you the best of success in conducting the review.

7. PLOS authors have the option to publish the peer review history of their article (what does this mean?). If published, this will include your full peer review and any attached files.

Reviewer #1: **Yes:** Mike Rommerskirch-Manietta

---

## [Editor Report · Acceptance letter]

PONE-D-25-60484R2

PLOS One

Dear Dr. Wong,

I'm pleased to inform you that your manuscript has been deemed suitable for publication in PLOS One. Congratulations! Your manuscript is now being handed over to our production team.

Kind regards,

on behalf of

Professor Sascha Köpke

Academic Editor

PLOS One